# Epigenetic aging differentially impacts breast cancer risk by self-reported race

Yanning Wu[1], Megan E. Miller[2,3], Hannah L. Gilmore[4], Cheryl L. Thompson[5‡], Fredrick R. Schumacher[1,6‡]*

1 Department of Population and Quantitative Health Sciences, School of Medicine, Case Western Reserve University, Cleveland, Ohio, United States of America, 2 University Hospitals Research in Surgical Outcomes and Effectiveness (UH-RISES), Cleveland, Ohio, United States of America, 3 Division of Surgical Oncology, Department of Surgery, University Hospitals Cleveland Medical Center, Case Western Reserve University School of Medicine, Cleveland, Ohio, United States of America, 4 Department of Pathology, Case Western Reserve University School of Medicine and University Hospitals Cleveland Medical Center, Cleveland, Ohio, United States of America, 5 Department of Public Health Sciences, Penn State College of Medicine, Hershey, Pennsylvania, United States of America, 6 Case Comprehensive Cancer Center, Case Western Reserve University School of Medicine, Cleveland, Ohio, United States of America

‡ CLT and FRS are joint authorship on this work.

* frs2@case.edu

## Abstract

### Background

Breast cancer (BrCa) is the most common cancer for women globally. BrCa incidence varies by age and differs between racial groups, with Black women having an earlier age of onset and higher mortality compared to White women. The underlying biological mechanisms of this disparity remain uncertain. Here, we address this knowledge gap by examining the association between overall epigenetic age acceleration and BrCa initiation as well as the mediating role of race.

### Results

We measured whole-genome methylation (866,238 CpGs) using the Illumina EPIC array in blood DNA extracted from 209 women recruited from University Hospitals Cleveland Medical Center. Overall and intrinsic epigenetic age acceleration was calculated–accounting for the estimated white blood cell distribution–using the second-generation biological clock GrimAge. After quality control, 149 BrCa patients and 42 disease-free controls remained. The overall chronological mean age at BrCa diagnosis was 57.4 ± 11.4 years and nearly one-third of BrCa cases were self-reported Black women (29.5%). When comparing BrCa cases to disease-free controls, GrimAge acceleration was 2.48 years greater (p-value = 0.0056), while intrinsic epigenetic age acceleration was 1.72 years higher (p-value = 0.026) for cases compared to controls. After adjusting for known BrCa risk factors, we observed BrCa risk increased by 14% [odds ratio (OR) = 1.14; 95% CI: 1.05, 1.25] for a one-year increase in GrimAge acceleration. The stratified analysis by self-reported race revealed differing ORs for GrimAge acceleration: White women (OR = 1.17; 95% CI: 1.03, 1.36), and Black women (OR = 1.08; 95% CI: 0.96, 1.23). However, our limited sample size failed to

**Data Availability Statement:** The data underlying this study cannot be shared publicly due to the lack of informed consent from participants. However, data are available from Case Western Reserve University and University Hospitals Cleveland

Medical Center Institutional Data Access /Ethics Committee for researchers who meet the criteria for access to confidential data. Interested researchers can contact Fredrick Schumacher at frs2@case.edu for further information. Additionally, data can be requested directly from OSF (https://osf.io/institutions/cwru) or by contacting a research data specialist at Kelvin Smith Library at CWRU (asksl@case.edu).

**Funding:** This study was funded by the National Cancer Institute (R03CA241956; P30CA043703). The funders had no role in study design, data collection and analysis, decision to publish, or preparation of the manuscript.

**Competing interests:** The authors have declared that no competing interests exist.

**Abbreviations:** BrCa, Breast cancer; DNAm, DNA methylation; GrimAA, GrimAge acceleration; IEAA, Intrinsic epigenetic age acceleration; OR, Odds ratio; CI, Confidence interval; UHCMC, University Hospitals Cleveland Medical Center.

detect a statistically significant interaction for self-reported race (*p-value* >0.05) when examining GrimAge acceleration with BrCa risk.

## Conclusions

Our study demonstrated that epigenetic age acceleration is associated with BrCa risk, and the association suggests variation by self-reported race. Although our sample size is limited, these results highlight a potential biological mechanism for BrCa risk and identifies a novel research area of BrCa health disparities requiring further inquiry.

## Background

According to the World Health Organization, breast cancer (BrCa) is the most common cancer for women worldwide, with over 2.3 million diagnoses in 2020 [1]. In the United States, BrCa is the most common malignancy and the second most lethal cancer for women [2], corresponding to one in every six cancer deaths in women [3]. In terms of morbidity and mortality, well-established disparities exist among different racial and ethnic groups [2, 3]. Although BrCa incidence is slightly lower for Black women compared to White women, with rates of 127.1 and 132.5 per 100,000 population, respectively, Black women have a 40% higher mortality rate compared with White women (28.0 vs. 19.9 per 100,000) [3]. Furthermore, Black women have an earlier age of onset compared to White women, and the incidence of BrCa before age 45 is higher for Black women compared to White women [2].

Regardless of race, several well-established risk factors are known to increase the likelihood of developing BrCa: aging [4], alcohol consumption [5], obesity [6], parity [7], and age at menarche [8]. Aging in general is a prominent risk factor for the majority of cancers, including BrCa [9, 10], and it is positively correlated with incidence. Seventy-seven percent of women diagnosed with BrCa are over the age of 50, whereas fewer than 1% are diagnosed in their 20s [11]. This large variation in BrCa incidence by age of onset has motivated cancer investigators to identify and better understand the underlying biological mechanisms of aging. For example, recent research has interrogated the association between decreasing telomere length, a hallmark of aging, and BrCa risk [12–17]. A series of epidemiological studies reported a significant negative correlation between telomere length and BrCa incidence [12–14]; however, subsequent studies failed to observe a significant association [15, 16], or even reported an inverse effect [17]. Additional biomarkers of aging are needed to assess associations with diseases, such as cancer.

Another growing area of research has focused on using DNA methylation (DNAm) aging as a marker of biological aging. DNAm aging is an epigenetic modification in which changes in DNA methylation patterns in different regions of the genome can regulate gene expression and are associated with the biological aging process [18]. The first-generation epigenetic clock was proposed over a decade ago, and utilized DNAm from multiple tissues to correlate DNAm age with chronological age [19]. Although the first-generation clocks provided valuable insights of aging and age-related diseases, several limitations have been observed. The development of second-generation epigenetic clocks (i.e. GrimAge clock, PhenoAge clock) utilize a larger panel of CpG sites and expanded their modeling beyond chronological aging to improve the accuracy and broader representation of aging associated DNAm changes, thus potentially predicting age-related health outcomes [20, 21].

Second-generation clocks have been studied to correlate with the incidence of all-cancers [20], as well as specific cancers [22–24]. Kresovich *et al.* observed that DNAm age acceleration, was significantly associated with increased risk of developing BrCa [23]. In this study, the authors reported that a five-year acceleration in DNAm age was associated with an 15% increase in odds of BrCa risk using the PhenoAge clock [23]. However, in a separate study, the findings showed no significant relationship between GrimAge acceleration (GrimAA) and overall BrCa incidence and only a weak positive association was observed between GrimAA and invasive BrCa [25].

Although most of the research to date on cancer and epigenetic aging has studied the association of overall epigenetic aging with cancer risk, emerging research on biological aging has suggested that changes in leukocytes play a crucial role in the aging process and have potential implications for the pathogenesis of age-associated diseases [26]. This has been defined as intrinsic epigenetic age acceleration (IEAA), which measures cell-intrinsic aging processes, adjusting for extra-cellular differences in leukocyte counts [27]. Population-based studies have revealed significant differences in IEAA between individuals with Parkinson's disease and controls [28].

However, the role of intrinsic aging acceleration on BrCa in human populations has not been well studied. The objectives of this study, therefore, are (1) to determine the association of age acceleration measures with BrCa risk, and (2) assess these associations stratified by self-reported race. We hypothesize BrCa cases have a greater biological age compared to disease-free controls, and this association is mediated by self-reported race.

## Results

### Descriptive statistics

We started with 209 individuals and then we excluded 15 duplicates plus 3 samples based on our QC pipeline. Table 1 provides descriptive statistics for our study population. More than one-third of our study participants were self-reported Black women (N = 65). The overall mean age of BrCa cases and disease-free controls was 57.4 ± 11.4 years and 54.4 ± 12.4 years, respectively. The majority of BrCa subtypes were ER+ (55.7%) and HER2- (79.9%). In addition, the average age at menarche, 12.7 years, was similar for both BrCa cases and disease-free controls. Over half of the BrCa cases were parous, 59.7% (N = 89), with at least one childbirth. Similarly, slightly more than half of the disease-free controls (54.8%) reported at least one childbirth. The average BMI was similar between BrCa cases and disease-free controls, 30.0 ± 7.89 vs 29.7 ± 8.62. The majority of BrCa cases were never smokers (54.4%), and slightly less than half of disease-free controls (45.2%) were never smokers. Although the missingness for parity and age at menarche was high, rates were similar by disease status. Among BrCa cases, pre-treated DNA was collected from 47 (31.5%) women and post-treated DNA from 102 (68.5%) women. Overall, BrCa treatment type was known for ~80% of DNA that was collected post-treatment. Among the post-treatment group, 55% of the women underwent chemotherapy or a combination of chemotherapy and radiotherapy. Post-treated DNA samples were collected in 8% of the women who underwent surgery alone and treatment status was missing for slightly over 20% of the women.

### Breast cancer cases had higher epigenetic age acceleration

We evaluated the association between the two age acceleration measures, GrimAA and IEAA (defined as the residual obtained by regressing GrimAge on chronological age adjusting for blood cell composition estimates), with BrCa. Using the GrimAge approach [20], we observed an increased mean GrimAA of 2.48 years for BrCa cases compared with disease-free controls

**Table 1. Demographic information of study population by breast cancer status.**

|  | Breast cancer cases (N = 149) | Disease-free controls (N = 42) |
|---|---|---|
| **Age** |  |  |
| Mean (SD) | 57.4 (11.4) | 54.4 (12.4) |
| **Self-reported Race** |  |  |
| Black | 44 (29.5%) | 21 (50.0%) |
| White | 105 (70.5%) | 21 (50.0%) |
| **BMI** |  |  |
| Mean (SD) | 30.0 (7.89) | 29.7 (8.62) |
| Missing | 7 (4.7%) | 4 (9.5%) |
| **Smoking Status** |  |  |
| Never | 81 (54.4%) | 19 (45.2%) |
| Former | 46 (30.9%) | 12 (28.6%) |
| Current | 16 (10.7%) | 5 (11.9%) |
| Missing | 6 (4.0%) | 6 (14.3%) |
| **Age at menarche (years)** |  |  |
| Mean (SD) | 12.7 (1.42) | 12.7 (1.62) |
| Missing | 47 (31.5%) | 17 (40.5%) |
| **Parity** |  |  |
| No | 13 (8.7%) | 5 (11.9%) |
| Yes | 89 (59.7%) | 23 (54.8%) |
| Missing | 47 (31.5%) | 14 (33.3%) |
| **ER** |  |  |
| Negative | 64 (43.0%) | NA |
| Positive | 83 (55.7%) | NA |
| Missing | 2 (1.3%) | NA |
| **HER2** |  |  |
| Negative | 119 (79.9%) | NA |
| Positive | 22 (14.8%) | NA |
| Missing | 8 (5.4%) | NA |
| **Tumor grade** |  |  |
| 1 | 12 (8.1%) | NA |
| 2 | 47 (31.5%) | NA |
| 3 | 69 (46.3%) | NA |
| Missing | 21 (14.1%) | NA |
| Treatment status at DNA collection |  |  |
| Pre-treatment | 47 (31.5%) | NA |
| Post-treatment | 102 (68.5%) | NA |
| Surgery only | 8 (7.8%) | NA |
| Chemotherapy only | 19 (18.6%) | NA |
| Radiotherapy only | 16 (15.7%) | NA |
| Both chemotherapy and radiotherapy | 37 (36.3%) | NA |
| Missing | 22 (21.6%) | NA |
| GrimAA |  |  |
| Mean (SD) | 0.59 (4.80) | -1.89 (5.93) |
| IEAA |  |  |
| Mean (SD) | 0.39 (4.22) | -1.33 (4.88) |

*SD* standard deviation; *BMI* Body mass index; *ER* estrogen receptor; *HER2* human epidermal growth factor receptor 2; *GrimAA* GrimAge acceleration; *IEAA* Intrinsic epigenetic age acceleration; *NA* not applicable

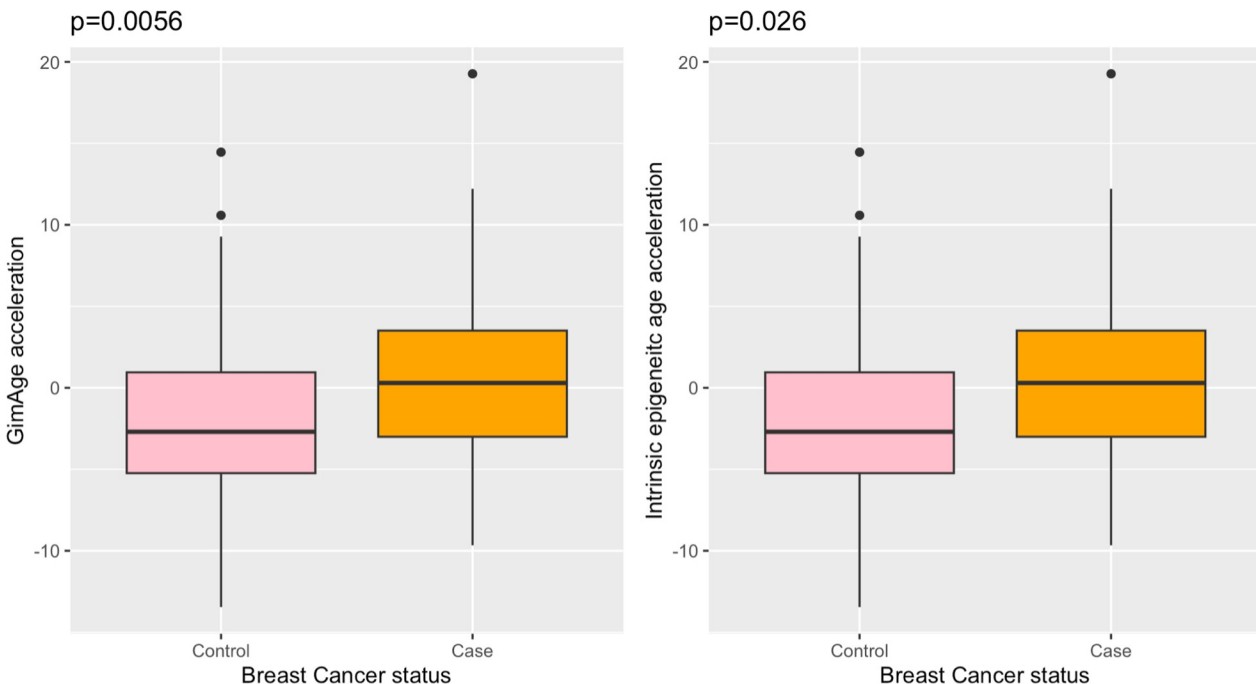

**Fig 1. Boxplots of different epigenetic age acceleration measures in breast cancer cases (orange) and disease-free controls (pink).** A. Boxplot of GrimAge acceleration between breast cancer cases and disease-free controls, B. Boxplot of intrinsic epigenetic age acceleration between breast cancer cases and disease-free controls.

(Fig 1A; *p*-value = 0.0056). Slightly attenuated, BrCa cases exhibited a significant 1.72-year higher IEAA compared with disease-free controls (Fig 1B; *p*-value = 0.026).

The results for our three logistic regression models are presented in Table 2. First, we performed a univariable logistic regression between age acceleration (GrimAA or IEAA) and BrCa status in Model 1. The result indicated that GrimAA was positively associated with BrCa risk (OR = 1.11; 95% CI: 1.03, 1.20; *p*-value = $6.82\times10^{-3}$). A similar effect size was observed for IEAA adjusted for leukocyte cell type, with an OR of 1.10, implying that each unit increase in IEAA could lead to a 10% increased odds of BrCa (*p*-value = 0.02; 95% CI: 1.01, 1.20).

Next, we included age and self-reported race in Model 2. Model 3 was further expanded to encompass variables in Model 2 as well as other known risk factors for BrCa (Table 2). Compared with the univariable model, the effect size of GrimAA tends to increase sequentially with the addition of BrCa explanatory variables: Model 2 OR = 1.13 (95% CI: 1.05, 1.23; *p*-value = $1.58\times10^{-4}$;) and Model 3 OR = 1.14 (95% CI: 1.05, 1.25; *p*-value = $2.86\times10^{-3}$). The

**Table 2. Logistic regression models by different epigenetic age acceleration measures.**

|  | Model 1[a] OR [95% CI] | P value | Model 2[b] OR [95% CI] | P value | Model 3[c] OR [95% CI] | P value |
|---|---|---|---|---|---|---|
| **GrimAA** | 1.11 [1.03, 1.20] | $6.82\times10^{-3}$ | 1.13 [1.05, 1.23] | $1.58\times10^{-4}$ | 1.14 [1.05, 1.25] | $2.86\times10^{-3}$ |
| **IEAA** | 1.10 [1.01, 1.20] | 0.02 | 1.14 [1.04, 1.25] | $4.50\times10^{-3}$ | 1.13 [1.03, 1.24] | 0.01 |

*GrimAA* GrimAge acceleration; *IEAA* Intrinsic epigenetic age acceleration; *OR* odds ratio; *CI* confidence interval

[a] Model 1 is the univariable logistic regression model between different epigenetic age acceleration measures and BrCa risk.

[b] Model 2 is the multivariable logistic regression model including age and self-reported race as covariates.

[c] Model 3 is the multivariable logistic regression model including age, self-reported race and other known BrCa risk factors (BMI, age at menarche, parity, smoking status) as covariates.

**Table 3. Multivariable logistics regressions stratified by self-reported race\*.**

|  | Black participants (N_Case = 44, N_Control = 21) | | White participants (N_Case = 105, N_Control = 21) | |
| --- | --- | --- | --- | --- |
|  | OR [95% CI] | P value | OR [95% CI] | P value |
| **GrimAA** | 1.08 [0.96, 1.23] | 0.22 | 1.17 [1.03, 1.36] | 0.02 |
| **IEAA** | 1.07 [0.94, 1.23] | 0.32 | 1.15 [1.01, 1.35] | 0.05 |

*GrimAA* GrimAge acceleration; *IEAA* Intrinsic epigenetic age acceleration; *OR* odds ratio; *CI* confidence interval
Note
\* Stratified analyzes were performed here using Model 3 in Table 2 (multivariable logistic regression models including age, self-reported race, and other known BrCa risk factors)

IEAA effect for Model 2 and Model 3 increased by 4% and 3%, respectively, compared to Model 1. However, the effect estimates for age acceleration from Model 2 and Model 3 were nearly identical (Table 2).

## Epigenetic age acceleration impacts BrCa risk by self-reported race

We then evaluated whether epigenetic age acceleration differs by self-reported racial groups in disease-free controls. Both self-reported racial groups had an estimated epigenetic age lower than chronological age in controls. However, disease-free Black women exhibited 2.79 years higher age acceleration in GrimAA on average compared with White women (S1A Fig, *p*-value = 0.13). Similarly, the mean IEAA was estimated as 2.57 years greater for Black women than for White women (S1B Fig, *p*-value = 0.09). These results imply that on average, the effect of age acceleration on BrCa risk when compared to disease-free controls is slightly greater among Black women compared to White women.

We further investigated the presence of an interaction effect between epigenetic age acceleration and self-reported race. The stratified analysis by self-reported race in Table 3 demonstrated that GrimAA had a significant association with BrCa risk in White women, with an OR of 1.17 (*p*-value = 0.02; 95% CI: 1.03, 1.36). Accounting for leukocyte components, the effect size for IEAA in White women decreased, yielding an OR of 1.15 (*p*-value = 0.05; 95% CI: 1.01, 1.35). In comparison to White women, the OR for Black women was observed to be smaller, 1.08 (95% CI: 0.96, 1.23). The stratified analysis revealed different ORs for IEAA based on self-reported race, however, the interaction term failed to reach statistical significance (*p*-value >0.05).

## Discussion

Using a retrospective case-control study design, our data suggests that: (1) both GrimAA and IEAA are positively associated with the risk of developing BrCa, and (2) self-reported race modifies the observed association.

GrimAA showed that BrCa cases aged, on average, 2.48-years faster than controls, and a one-year increase in GrimAA was associated with a 14% increased risk of BrCa, after adjusting for potential BrCa risk factors. More specifically, the risk of developing BrCa is 1.48 with a three-year increase in GrimAA. Likewise, a three-year increase in IEAA yields an OR of 1.44 for a woman developing BrCa.

GrimAA consistently showed a statistically significant association with BrCa in both the unadjusted and adjusted analyses. This finding is not surprising, partially because GrimAge clock is constructed on 7 plasma proteins, e.g. leptin and cystatin C, and smoking pack-years [20]. Among plasma proteins, plasma leptin levels have been shown to be a reliable biomarker

of BrCa risk in premenopausal women [29, 30]. Further, previous studies demonstrated similar findings [23, 31, 32]. A recent study reported that DNAm predictors of leptin and cystatin C were strongly associated with BrCa incidence even after correcting for BrCa risk factors [31]. Our study further validated that DNAm predictors of cystatin C (*p*-value = 0.03) and leptin (*p*-value = 0.007) remained strongly associated with BrCa incidence adjusting for known BrCa risk factors and white blood cell compositions. Therefore, the significant association between BrCa and GrimAA may partly be due to the inclusion of CpGs near these plasma proteins, thereby capturing a causal mechanism closer to BrCa initiation.

In addition to GrimAA, we also observed significant findings in IEAA with BrCa. IEAA is derived from GrimAge clock accounting for white blood cell composition, thus mitigating the impact of cell composition on epigenetic age estimates. IEAA demonstrated a similar significant result compared to GrimAA, albeit with a slightly attenuated effect size. Kresovich et al. [33] previously reported that circulating leukocyte profiles may serve as a marker of BrCa risk. This insight helps in elucidating that the attenuated effect size in our study suggests that biological age and blood cell composition may both be signals of BrCa risk, potentially due to a shared or overlapping pathway.

We evaluated the robustness of our associations by comparing the mean age acceleration between the DNA collections pre-treatment and post-treatment using the Wilcoxon test. The results, shown in S2 Fig, indicate significant differences in GrimAA (p < 0.001), but no significant differences in IEAA (p = 0.80) between the two groups. In S1 Table, the ORs from Model 2 range from 1.09–1.13 for IEAA by BrCa treatments on post-treatment collected DNA compared to disease-free controls. This range includes the OR effect observed in the pre-treatment collected DNA (OR = 1.11) compared to disease-free controls, this demonstrating a minimal impact. This observation has been previously published literature [34, 35], suggesting that IEAA may be less sensitive to unaccounted factors, thereby mitigating the effects of residual confounding. However, nearly 20% of BrCa treatment information was missing from women who had their DNA collected post-treatment, and this may affect our results.

We also examined the association between BrCa risk and other commonly used methylation clocks, such as the Horvath clock [19] and PhenoAge clock [21], as well as DNAm-based metrics of aging rates [36]. The results are presented in S1 Table. PhenoAge acceleration yielded a similar direction to our reported GrimAge acceleration, likely due to its similar objective of predicting mortality and incorporation of clinical measures in the model development. However, the Horvath clock displayed an opposite direction of association, possibly attributable to its design specifically for predicting chronological age. The DNAm-based metrics of aging rates (DunedinPACE), often referred to as the "third-generation clock", serve as a biomarker measuring the human pace of aging [36] and exhibited a similar and significant association with BrCa risk (OR = 1.37; 95% CI: 1.08, 1.79; *p*-value = 0.014). It's important to note that while we reported associations measured by different epigenetic clocks, the results are not directly comparable, as each clock was developed differently with distinct objectives.

In a supplementary analysis of disease-free controls presented in S1 Fig, our data suggested that in our control cohort, the biological GrimAge is consistently lower than the chronological age for both Black and White individuals, with a more pronounced difference observed in White women compared to Black women. This finding is interesting compared to our interaction model, suggesting that age acceleration measures have a greater protective effect at baseline among individuals of self-reported European ancestry compared to self-reported African ancestry. However, when an individual develops BrCa, this effect became more pronounced in self-reported White individuals. In our sample, Black women age faster than White women using both epigenetic clocks. However, likely due to the limited sample size, the results failed to reach statistical significance. While prior studies have shown similar findings, earlier results

were generated from populations consisting of both males and females [20, 29]. Tajuddin et al. conducted a prospective longitudinal study by selecting middle-age African American and White participants below poverty status, and concluded that African American individuals exhibited faster intrinsic aging compared to White individuals, although the observed difference did not reach statistical significance [37]. In addition, recent findings suggested that 110 of the 353 CpGs in the Horvath clock and neighboring stress-responsive genes can be regulated by glucocorticoid receptor activation, and showed an enriched association with aging-related diseases in African American individuals [38]. Although widespread data on CpGs for the GrimAge clock is lacking, the DNAm clock shows promise as a biomarker that could potentially connect health disparities and disease risk factors, particularly in BrCa.

Our study is unique due to the racially diverse study population. A majority of previous studies were conducted among individuals of European decent [25, 39], and cannot be extrapolated to non-European populations. Our study helps to address this scientific gap. In addition, we applied the latest epigenome wide technology (Infinium MethylationEPIC array), and chose multiple measurements of epigenetic aging acceleration, including universal and intrinsic measures. We also tested the interaction effect for age acceleration and self-reported race in modifying BrCa risk, an important and sometimes overlooked relationship. However, our study has several limitations. The moderate size of our study population must be taken under consideration when interpreting our results and our limited analysis of individual hormone receptors due to minimal statistical power. Furthermore, we relied on self-reported race rather than estimated genetic/continental ancestry, thus potentially leading to misclassification of race at the individual level. These are areas of focus in future analyses.

## Conclusions

In summary, our study demonstrated that epigenetic age acceleration is positively associated with BrCa risk. An increased GrimAA and IEAA are both associated with the risk of BrCa, after adjusting for potential confounding risk factors. The stratification analysis by self-reported race indicated differences in effect measures, however a test for interaction failed to reach statistical significance ($p$-value >0.05). This suggests that varied DNAm patterns among self-reported racial groups may be important in age-related BrCa health disparities. Although our sample size is moderately sized, our results provide evidence of a novel research direction in cancer disparities research. Additional studies with larger sample sizes are needed to replicate our findings.

## Materials and methods

### Study population

The present study is a subset of a case-control study including 209 women recruited from University Hospitals Cleveland Medical Center (UHCMC) between 2007 and 2019, including 160 BrCa cases and 49 disease-free controls. BrCa cases were patients diagnosed with BrCa at UHCMC and disease-free controls were mammography screened with no previous history of cancer. At recruitment, all participants completed a survey and provided a blood sample. Clinical and long-term follow-up data was collected for each participant from medical records. Demographics including self-reported race and lifestyle variables were self-reported through a survey at recruitment. The molecular subtypes were abstracted from medical records. Human subjects research was approved by the Institutional Review Board at UHCMC (Protocol #CASE3116) and written consent was obtained for all study participants. Annotated research databases were accessed starting on 10/23/2022 and currently continues.

## DNA methylation measure

Genomic DNA was quantified by Qubit Fluorometer (Invitrogen) and qualified by agarose gel and processed for methylation assays. The Infinium MethylationEPIC BeadChip (Illumina) is a comprehensive array that covers over 850,000 methylation sites quantitatively across the genome at single-nucleotide resolution following the Infinium HD assay (Illumina). An input of 250ng of gDNA was used for the bisulfite conversion using the EZ DNA Methylation Kit (Zymo). Once bisulfite conversion was complete, the Infinium HD protocol was followed using the MethylationEPIC BeadChip. First, the DNA was amplified, fragmented, and precipitated to prepare for hybridization to the Beadchip. Eight samples can be loaded onto one Beadchip. Single-base extension of the oligos on the BeadChip was performed on the hybridized DNA. Beadchips were then scanned on the Illumina iScan system. All raw data, along with the corresponding manifest file, decode file, and sample sheet were uploaded into Illumina's GenomeStudio Methylation software module. Internal controls were checked within the software to validate complete bisulfite conversion and successful processing of the Infinium HD assay for each sample. Data was processed for downstream analysis. The DNAm level was estimated as β values, ranging from 0 (unmethylated) to 1 (fully methylated) for each probe.

A quality control pipeline was performed using the *minfi* package in R 4.3.2. In brief, probes were removed with a detection p-value, which is always used for assessing individual probe performance, greater than 0.05 in more than 5% of all samples [40]. Samples with a mean detection p-value greater than 0.05 were excluded. The cross-reactive probes and probes overlapping with known SNPs were also excluded. We then performed background correction, normalization, and batch effect adjustment. After all processes, three samples were dropped from the study population, and β values for 807,893 methylation probes were obtained.

## Epigenetic clock measurement

Both GrimAA and IEAA was calculated using the online DNAm age calculator [20] (https://dnamage.genetics.ucla.edu/home). In order to quantify aging acceleration, we defined GrimAA as the residual from regressing chronological age over methylation GrimAge. A positive GrimAA indicated the study participant was biologically older than their chronological age indicated, whereas a negative GrimAA identified a study participant who was biologically younger than their chronological age at the time of blood draw. Our DNAm estimation accounted for the influence of cellular heterogeneity by adjusting for measures of cell abundance. In our analysis, we included the following estimated white blood cell compositions: naive CD8+ T cells, exhausted CD8+ T cells, plasmablasts, CD4+ T cells, natural killer cells, monocytes, and granulocytes [27]. The abundance of naive CD8+ T cells, exhausted CD8+ T cells, plasmablasts was determined using a previously published approach [41], while the remaining cell types were imputed using a different approach [42]. IEAA measures were derived to eliminate the confounding effect of cell abundance. IEAA was defined as a residual of regressing GrimAge on chronological age and adjusting for white blood cell compositions.

## Statistical analysis

We performed statistical analysis via R 4.3.2. The association between age acceleration and BrCa status was evaluated by Student's t-test. OR for BrCa risk and 95% CI were calculated using unconditional logistic regression, where the dependent variable was BrCa status (BrCa case vs cancer-free control) and the independent variable was an epigenetic age acceleration measure. Both univariable and multivariable logistic regression models were conducted. The univariable Model 1 included the epigenetic age acceleration variable alone. The multivariable model of epigenetic age acceleration additionally adjusted for self-reported race (Black/White)

in Model 2 and multiple BrCa risk factors [self-reported race (Black/ White), BMI (continuous), age at menarche (continuous), parity (Yes/ No) and smoking status (Never/ Former/ Current)] were added to the epigenetic age acceleration in Model 3. To maintain a comparable sample size for Model 3, missing data were imputed using the mean value for continuous variable (*e.g.* age at menarche) stratified by self-reported race using linear regression. Categorical variables (*e.g.* smoking status and parity) missing values were imputed by assigning them to an independent category. A stratified analysis based on self-reported race was performed using different age acceleration measures, also adjusting for multiple BrCa risk factors. The interaction model was carried out by adding an interaction term between self-reported race and age acceleration measures in Model 3. Statistical significance was defined as a p-value <0.05, and all statistical tests were two-sided.

## Supporting information

**S1 Fig. Boxplots of different epigenetic age acceleration measures of disease-free controls by race.** A. Boxplot of GrimAge acceleration in disease-free controls between Black and White, B. Boxplot of intrinsic epigenetic age acceleration in disease-free controls between Black and White.
(TIF)

**S2 Fig. Boxplots of different epigenetic age acceleration measures by treatment status at DNA collection.** A. Boxplot of GrimAge acceleration between the DNA collections pre-treatment and post-treatment, B. Boxplot of intrinsic epigenetic age acceleration between the DNA collections pre-treatment and post-treatment.
(TIF)

**S1 Table. The effect between different types of treatments on various epigenetic age acceleration measures.**
(DOCX)

**S2 Table. The associations between multiple epigenetic clocks and breast cancer risk.**
(DOCX)

## Acknowledgments

The Genomics Core of the Case Comprehensive Cancer Center provided expertise and input on sample preparation and quality control.

## Author Contributions

**Conceptualization:** Cheryl L. Thompson, Fredrick R. Schumacher.

**Data curation:** Megan E. Miller, Hannah L. Gilmore, Cheryl L. Thompson, Fredrick R. Schumacher.

**Formal analysis:** Yanning Wu.

**Funding acquisition:** Cheryl L. Thompson, Fredrick R. Schumacher.

**Methodology:** Yanning Wu.

**Supervision:** Fredrick R. Schumacher.

**Visualization:** Yanning Wu.

**Writing – original draft:** Yanning Wu, Cheryl L. Thompson, Fredrick R. Schumacher.

**Writing – review & editing:** Yanning Wu, Megan E. Miller, Hannah L. Gilmore, Cheryl L. Thompson, Fredrick R. Schumacher.

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
