## [Decision Letter · Decision Letter 0]

9 Jan 2024

PONE-D-23-31393Epigenetic aging differentially impacts breast cancer risk by self-reported ancestryPLOS ONE

Dear Dr. Schumacher,

Thank you for submitting your manuscript to PLOS ONE. After careful consideration, we feel that it has merit but does not fully meet PLOS ONE’s publication criteria as it currently stands. Therefore, we invite you to submit a revised version of the manuscript that addresses the points raised during the review process.

We look forward to receiving your revised manuscript.

Kind regards,

Abdul Rauf Shakoori

Academic Editor

PLOS ONE

Journal Requirements:

"This study was funded by the National Cancer Institute (R03CA241956; P30CA043703)."

5. In this instance it seems there may be acceptable restrictions in place that prevent the public sharing of your minimal data. However, in line with our goal of ensuring long-term data availability to all interested researchers, PLOS’ Data Policy states that authors cannot be the sole named individuals responsible for ensuring data access (http://journals.plos.org/plosone/s/data-availability#loc-acceptable-data-sharing-methods).

6. We notice that your supplementary figure S1 is included in the manuscript file. Please remove them and upload them with the file type 'Supporting Information'. Please ensure that each Supporting Information file has a legend listed in the manuscript after the references list.

Reviewers' comments:

Reviewer's Responses to Questions

**Comments to the Author**

1. Is the manuscript technically sound, and do the data support the conclusions?

Reviewer #1: Yes

Reviewer #2: Yes

2. Has the statistical analysis been performed appropriately and rigorously? 

Reviewer #1: Yes

Reviewer #2: Yes

3. Have the authors made all data underlying the findings in their manuscript fully available?

Reviewer #1: Yes

Reviewer #2: No

4. Is the manuscript presented in an intelligible fashion and written in standard English?

Reviewer #1: Yes

Reviewer #2: Yes

5. Review Comments to the Author

Reviewer #1: In the manuscript “Epigenetic aging differentially impacts breast cancer risk by self-reported ancestry,” Wu et al. perform a case-control study of newly-diagnosed breast cancer patients and cancer free controls with two DNA methylation (DNAm) based metrics of biological age. The investigators report positive associations between age acceleration metrics (both adjusted an unadjusted for blood cell composition) and breast cancer risk. In a stratified analysis by self-reported race, the association between age acceleration and breast cancer risk appeared to vary, with stronger associations observed in White women. Overall, the paper is a bit sophomoric and narrow in focus. However, it is well written and advances our understanding of the association between the GrimAge epigenetic clock and breast cancer using a small, but racially diverse, population. Although I believe the manuscript is worthy of publication in this journal, the authors should address the following points:

Major concerns

There seems to be some confusion about the role of self-reported race in the analysis of GrimAge and breast cancer. The results from the stratified models and the significant interaction term suggest that race is a modifier of the association, not a confounder or a mediator as suggested by the authors.

It is perhaps not surprising that the association of intrinsic GrimAge and breast cancer is attenuated compared to the unadjusted model; Kresovich et al (2020, JAMA Network open) previously reported that blood cell composition is a marker of breast cancer risk suggesting that biological age and blood cell composition appear to detect overlapping, although not entirely, signals of breast cancer risk. This point should be made clear to readers.

The authors should report breast cancer associations with the DNAm proteins that comprise the GrimAge epigenetic clock. This would provide more of a basis to the comments made in the discussion where the authors suggest their associations are due to the underlying associations with cystatin C and leptin. (see Kresovich et al, Aging 2019)

Although using the MethylationEPIC BeadChip information for biological age is interesting, there is a bit of a missed opportunity to perform an epigenome-wide association study of breast cancer at the individual CpG site resolution. A challenge would be replicating associations, but many EWASs of breast cancer have been performed, so the investigators could present an external validation of known CpG and BC associations.

Similarly, the associations between other epigenetic clocks (Horvath, PhenoAge) and DNAm-based metrics of aging rates (DunedinPACE) should be at a minimum reported in supplemental material. There do not appear to be any papers between aging rates (DunedinPACE) and breast cancer, which would greatly strengthen the impact of this manuscript.

Minor concerns

There is some incorrect information about GrimAge in the discussion: the GrimAge clock is based on 7 DNAm-predicted plasma protein concentrations (not 88) and a DNAm predictor of smoking status, combined with chronological age and sex. The authors may need to revisit the original clock manuscripts to make sure their descriptions of the development procedures are correct.

The intrinsic GrimAge metric does not consider “white blood cell count” (line 200); instead, it accounts for “white blood cell composition” which is represented by the circulating proportion of different leukocyte subsets. Unfortunately, it is not possible to obtain reliable cell count information from methylation arrays.

The description of the figures seems to be incorrect. Please revise the figure images so the cases are pink and the disease-free controls are orange (opposite to what is noted in the figure legend).

Please do not confuse self-reported race with ancestry. Ancestry is commonly measured using genetic data whereas race is often based on identity. Unless genetic data were used to inform the ancestry of the participants, please refer to the data as how the participants “self-identified” themselves.

Reviewer #2: Yanning Wu and colleagues analyzed epigenetic clock data from 209 women of the Cleveland Medical Center with respect to an association with breast cancer. Statistically significant differences in age acceleration were found between cases and controls and this association persisted after adjustment for known confounder in a logistic regression model. The role of self-reported race was analyzed in subgroup analyses.

This paper is an interesting and important contribution to the field and the available data as well as the statistical analyses employed seem fit to answer the author’s research question. The study is well designed and the data seems to be of a high scientific quality. However, some issues remain and I hope the authors will find my comments helpful.

First, I found the nomenclature sometimes confusing. I would suggest using the convention in the field as this would make it much easier to comprehend the information presented. Namely, the abbreviation AA_GrimAge and AA_GrimAge_In seem weird to me. In most studies you would find the following abbreviations:

- DNA methylation age: DNAmA

- DNA methylation age acceleration: DNAmAA

- GrimAge acceleration: GrimAA

- Intrinsic epigenetic age acceleration: IEAA

- Extrinsic epigenetic age acceleration: EEAA

Also, it is necessary to distinct better between methylation age and methylation age acceleration. Both are used interchangeably in the manuscript which is not correct (methylation age is the “raw” biological age measure and methylation age acceleration refers to the “gap” between chronological and biological age).

Second, in some parts of the manuscripts the authors use causal language although causality is hard to deduct from this kind of (cross-section and observational) data. To confidently assume a causal effect, much more analyses and theoretical considerations need to be included (directed acyclic graph and others). Therefore, I would suggest rewriting the respective passages, e.g. exchange “results in” with “is associated with”.

Third, although it is understandable that the authors focus on GrimAge in this manuscript, it still would be interesting to see how the other epigenetic clocks (e.g. Horvath, Hannum, PhenoAge, DundinPACE) are associated with breast cancer. Additionally, this would increase comparability to other studies and substantially improve the impact of the study. I therefore suggest the authors provide these additional analyses as Supplementary Material and briefly discuss potential differences in the results. Since the authors used Steve Horvath’s website the other clocks should already be available.

Minor concerns:

Line 39: I think the convention is to use “intrinsic epigenetic age acceleration” when referring to the cell-type adjusted residuals. “GrimAge intrinsic clocks” sounds unfamiliar to me.

Line 44/45 “GrimAge intrinsic accelearion”: The same applies here. I think it would be better to stick to “intrinsic epigenetic age acceleration”.

Line 101: “intrinsic epigenetic age”: I think it should state “intrinsic epigenetic age acceleration”.

Line 103 “intrinsic epigenetic age” -> “intrinsic epigenetic age acceleration”

Line 106: “intrinsic age” -> “intrinsic age acceleration”

Line 137: Since you calculated a regression please exchange “correlated” with “associated”.

Line 138-140: I find this sentence confusing. The measure AA_Grim_In is per definition an cell-type adjusted residual of a regression of epigenetic age on chronological age. It therefore would not make sense to adjust any regression including AA_Grim_in for blood cell composition. I therefore would suggest that this sentence is rephrased, e.g. “similar effect size was found for the leukocyte cell type adjusted AA_Grim_In as well”.

Line 142: The word “impact” implies causation as well as a direction of the causal association. Neither can be deducted from cross-sectional data only. Please rephrase. Also this sentence is somewhat redundant to the second sentence in this paragraph.

Line 145: “effect” implies causation. Additionally, a potential effect of one variable on the other stays always the same. Only the effect size estimated from a statistical analysis changes.

Line 148 and 149: Please write “percentage points” instead of “%” if you are referring to the numerical difference between the risk.

Line 158: I am not sure whether self-reported race would “confound” the association. It certainly is as effect measure modifier (EMM) but whether if fulfils the criteria for a confounder should be checked.

Discussion: It is a frequently observed phenomenon that women have lower epigenetic age than chronological age (while men often have a higher epigenetic age). This is described frequently in the literature and might be important to mention in the discussion as this might not be known to all readers. Without context it could be confusing that the disease-free women are all epigenetically younger.

Line 294: What do you mean by “imputed”? Were they measured and the missing values were imputed? Or were they estimated from the methylation data? Please clarify.

Line 274: “Data was processed for downstream analyses”: Please elaborate which quality control measures were used and how data was processed (software, packages).

Statistical Analysis: Please report which statistical software was used to conduct the analyses.

Figure 1 and S1: Please use boxplots instead of dynamite plots as a lot of information is lost. If possible, please show individual datapoints to illustrate the distribution.

Table 1: Please include the epigenetic age estimations in your table one (AA_GrimAge and AA_GrimAge_In).

Table 3: Which of the models in Table 2 was used here? I think it would be interesting to show all three models of Table 2 also stratified by self-reported race.

List of abbreviations: IEAA and EEAA are noted but never used in the manuscript (although I would suggest to use these abbreviations instead of AA_GrimAge and AA_GrimAge_In).

6. PLOS authors have the option to publish the peer review history of their article (what does this mean?). If published, this will include your full peer review and any attached files.

Reviewer #1: No

Reviewer #2: **Yes: **Valentin Vetter

---

## [Author Response · Author response to Decision Letter 0]

24 Mar 2024

Please see attached document - "Response to Reviewer Critiques"

---

## [Decision Letter · Decision Letter 1]

19 Apr 2024

PONE-D-23-31393R1Epigenetic aging differentially impacts breast cancer risk by self-reported racePLOS ONE

Dear Dr. Schumacher,

Thank you for submitting your manuscript to PLOS ONE. After careful consideration, we feel that it has merit but does not fully meet PLOS ONE’s publication criteria as it currently stands. Therefore, we invite you to submit a revised version of the manuscript that addresses the points raised during the review process.

here is one issue with the analysis that has not been addressed. The differences in age acceleration being greater in the breast cancer patients appears real but the authors must take into account the effect of chemotherapy on age acceleration as this would skew any results in favor of greater age acceleration which is not due to breast cancer per se. So, in the analysis this must be controlled for.

We look forward to receiving your revised manuscript.

Kind regards,

Abdul Rauf Shakoori

Academic Editor

PLOS ONE

Journal Requirements:

Reviewers' comments:

Reviewer's Responses to Questions

**Comments to the Author**

1. If the authors have adequately addressed your comments raised in a previous round of review and you feel that this manuscript is now acceptable for publication, you may indicate that here to bypass the “Comments to the Author” section, enter your conflict of interest statement in the “Confidential to Editor” section, and submit your "Accept" recommendation.

Reviewer #3: (No Response)

2. Is the manuscript technically sound, and do the data support the conclusions?

Reviewer #3: Yes

3. Has the statistical analysis been performed appropriately and rigorously? 

Reviewer #3: Yes

4. Have the authors made all data underlying the findings in their manuscript fully available?

Reviewer #3: Yes

5. Is the manuscript presented in an intelligible fashion and written in standard English?

Reviewer #3: Yes

6. Review Comments to the Author

Reviewer #3: There is one issue with the analysis that has not been addressed. The differences in age acceleration being greater in the breast cancer patients appears real but the authors must take into account the effect of chemotherapy on age acceleration as this would skew any results in favor of greater age acceleration which is not due to breast cancer per se. So, in the analysis this must be controlled for.

7. PLOS authors have the option to publish the peer review history of their article (what does this mean?). If published, this will include your full peer review and any attached files.

Reviewer #3: No

---

## [Author Response · Author response to Decision Letter 1]

16 Jul 2024

Response to Reviewers

We thank the reviewers and editors for their time and consideration of our manuscript. Please find our response to the critiques and comments provided.

Reviewer #3: There is one issue with the analysis that has not been addressed. The differences in age acceleration being greater in the breast cancer patients appears real but the authors must take into account the effect of chemotherapy on age acceleration as this would skew any results in favor of greater age acceleration which is not due to breast cancer per se. So, in the analysis this must be controlled for.s

Response: Thank you for this comment. We have included several additional analyses addressing this important concern and strengthening our manuscript. 

In the revised manuscript, breast cancer treatment status at DNA collection was added to the text and Table 1 (excerpt; data added). Among breast cancer cases, pre-treated DNA was collected from 47 (31.5%) women and post-treated DNA from 102 (68.5%) women. The distribution of treatment types for DNA collected post-treatment is presented in Table 1. Treatment type was known for ~80% of DNA collected post-treatment. A DNA sample was collected following chemotherapy or a combination of chemotherapy and radiotherapy in approximately 55% of the women. Post-treated DNA samples were collected in 8% of the women who underwent surgery only, whereas treatment status was missing for slightly over 20% of the women. The missing treatment data may impact our results. We have added this to the discussion. 

Table 1 (new data):

Treatment status at DNA collection Breast cancer cases Disease-free controls

 Pre-treatment 47 (31.5%) NA

 Post-treatment 102 (68.5%) NA

 Surgery only 8 (7.8%) NA

 Chemotherapy only 19 (18.6%) NA

 Radiotherapy only 16 (15.7%) NA

 Both chemotherapy and radiotherapy 37 (36.3%) NA

 Missing 22 (21.6%) NA

To assess the impact of breast cancer treatment on the association of age acceleration (AA) measures, we compared the mean AA by treatment status at DNA collection using the Wilcoxon test. The results, shown in the following figure, indicate post-treatment GrimAge acceleration (GrimAA) was 1.79 units greater than pre-treated GrimAA (p = 0.003), however no mean difference was observed between the post- and pre-treatment groups for intrinsic epigenetic age acceleration (IEAA) (p=0.80). 

Based on these results, our data suggests that breast cancer treatment has less impact on IEAA. This result demonstrates the robustness of our conclusions and has been observed in the published literature1-2, thus suggesting that IEAA may be less sensitive to unaccounted factors, thereby mitigating the effects of residual confounding. We have added this statement to the discussion section, as indicated on line 439. And the above figure will be named as S2 Fig and added to the manuscript.

We also compared the effect of different types of treatments on GrimAA and IEAA among those patients whose samples were collected post-treatment. We used a different subset of breast cancer cases by different types of treatments and compared them to the same set of disease-free controls to assess effects (ORs) across treatment groups. Multivariable logistic regression models were conducted, adjusting for age and self-reported race (Model 2). The results were shown in the Table below and have also been added to the manuscript as S1 Table.

 OR(GrimAA) Pvalue OR(IEAA) Pvalue

Post-treatment Surgery only cases (N=8)

 vs controls (N=49) 1.23 0.03 1.13 0.03

 Chemotherapy only cases (N=19)

vs controls (N=49) 1.17 0.03 1.11 0.18

 Radiotherapy only cases (N=16)

vs controls (N=49) 1.12 0.07 1.09 0.27

 Chemotherapy and radiotherapy cases (N=37)

vs controls (N=49) 1.23 4x10-4 1.13 0.03

Pre-treatment Pre-treatment cases (N=47)

vs controls (N=49) 1.05 0.20 1.11 0.03

Overall, the IEAA OR is very consistent across all groups (range: 1.09-1.13), further suggesting that treatment had minimal impact on DNA collected following treatment. It is worth noting, the GrimAA ORs had more variation and ranged from 1.05-1.23. Most likely this is an artifact of our limited sample size. 

Here, we limited our analysis to Model 2 adjusting for age and self-reported race based on our limited sample size. Furthermore, the effect estimates from Model 2 presented in Table 2 with pre- and post-treated DNA samples combined were nearly identical, indicating either very minimal or no confounding from including additional breast cancer risk factors.

Citations

1. Brägelmann, Johannes, and Justo Lorenzo Bermejo. "A comparative analysis of cell-type adjustment methods for epigenome-wide association studies based on simulated and real data sets." Briefings in Bioinformatics 20.6 (2019): 2055-2065.

2. Pottinger, Tess D., et al. "Association of cardiovascular health and epigenetic age acceleration." Clinical epigenetics 13 (2021): 1-6.

---

## [Editor Report · Decision Letter 2]

18 Jul 2024

Epigenetic aging differentially impacts breast cancer risk by self-reported race

PONE-D-23-31393R2

Dear Dr. Schumacher,

We’re pleased to inform you that your manuscript has been judged scientifically suitable for publication and will be formally accepted for publication once it meets all outstanding technical requirements.

Kind regards,

Abdul Rauf Shakoori

Academic Editor

PLOS ONE
---

## [Editor Report · Acceptance letter]

2 Aug 2024

PONE-D-23-31393R2 

PLOS ONE

Dear Dr. Schumacher, 

I'm pleased to inform you that your manuscript has been deemed suitable for publication in PLOS ONE. Congratulations! Your manuscript is now being handed over to our production team.

Kind regards, 

on behalf of

Dr. Abdul Rauf Shakoori 

Academic Editor

PLOS ONE